# Using Time Lapse Monitoring for Determination of Morphological Defect Frequency in Feline Embryos after in Vitro Fertilization (IVF)

**DOI:** 10.3390/ani10010003

**Published:** 2019-12-18

**Authors:** Barbara Kij, Joanna Kochan, Agnieszka Nowak, Wojciech Niżański, Sylwia Prochowska, Karolina Fryc, Monika Bugno-Poniewierska

**Affiliations:** 1Department of Animal Reproduction, Anatomy and Genomics, University of Agriculture, Mickiewicza 24/28, 30-059 Krakow, Poland; barbara.kij@op.pl (B.K.); joanna.kochan@urk.edu.pl (J.K.); nowak.a.a@gmail.com (A.N.); 2Department of Reproduction and Clinic of Farm Animals, University of Environmental Science, Grundwaldzki square 49, 50-357 Wroclaw, Poland; wojciech.nizanski@upwr.edu.pl (W.N.); sylwia.prochowska@upwr.edu.pl (S.P.); 3Department of Animal Nutrition, Biotechnology and Fishering, University of Agriculture, Mickiewicza 24/28, 30-059 Krakow, Poland; fryc.fryc@gmail.com

**Keywords:** cat, embryos, morphological disorders, time-lapse

## Abstract

**Simple Summary:**

This study was conducted with the aim of determining the frequency of morphological defects in feline embryos, their competence to reach the blastocyst stage, and their ability to hatch. Embryonic morphological disorders affect development potential, and the use of time lapse monitoring (TLM) guarantees the precise observation of any changes that occur during in vitro embryo development.

**Abstract:**

Some human, bovine, and mouse in vitro fertilized (IVF) embryos with morphokinetic abnormalities such as fragmentation, direct cleavage, and cytoplasmic vacuoles have the potential to reach the blastocyst stage, which is related to a high potential for implantation. The latest techniques of embryo development observation to enable the evaluation and selection of embryos are based on time lapse monitoring (TLM). The aim of this study was to determine the frequency of morphological defects in feline embryos, their competence to reach the blastocyst stage, and their ability to hatch. Oocyte-cumulus complexes were isolated after the scarification of ovaries and matured in vitro. Matured oocytes were fertilized in vitro by capacitated spermatozoa. Randomly selected oocytes were observed by TLM for seven-to-eight days. Out of 76 developed embryos, 41 were morphologically normal, of which 15 reached the blastocyst stage. Of 35 abnormally developed embryos, 17 reached the blastocyst stage, of which six had single aberrations and 11 had multiple aberrations. The hatching rate (%) was 15.6% in normally cleaving embryos, 6.25% in embryos with single aberrations, and 3.33% in those with multiple aberrations. The present study reports the first results, found by using TLM, about the frequency of the morphological defects of feline embryos, their competence to reach the blastocyst stage, and their ability to hatch.

## 1. Introduction

According to the literature, some human, bovine, and mouse in vitro fertilized (IVF) embryos with morphokinetic abnormalities, such as fragmentation and direct cleavage, from one-to-three or two-to-five blastomeres or cytoplasmic vacuoles are able to reach the blastocyst stage. It should be emphasized that this stage is considered to have a high potential for implantation [1,2,3]. The latest techniques for observing embryo development are based on time lapse monitoring (TLM). The use of TLM enables the evaluation and selection of those embryos with the highest potential for development, which has a positive effect on efficiency of in vitro fertilization and, as a consequence, the pregnancy rate [4,5]. According to the literature, TLM can distinguish morphological aberrations like the fragmentation of the cytoplasm [4], cytoplasmic vacuoles [6], and direct cleavage [7]. Fragmentation is the most often occurring abnormality that is exhibited among human embryos after in vitro fertilization [4,8]. The degree of fragmentation is one of the parameters that is necessary to determine the quality of human embryos. Embryo cleavage from one-to-three blastomeres all at once or from two-to-three blastomeres less than five hours between divisions is defined as “direct cleavage” [7,9]. Embryos with this aberration may develop better, but the pregnancy rate from such embryos is lower than with normally cleaving embryos [10,11]. Another morphological aberration that can be observed in embryos is that of vacuoles. Their presence is associated with reduced fertilization rates and embryo developmental potential [12,13].

IVF is a useful technique when applied to conservation of endangered cats [14]. Due to the low efficiency of IVF in cats and because of the critically low populations of wild felids, it is important to quantify the frequency of the abnormal development of cat embryos. TLM has been used to observe mouse and cattle embryos, but there are no data about the TLM observation of cat embryos [3,7]. The aim of the study was to determine the frequency of morphological defects of feline embryos, their competence to reach the blastocyst stage, and their ability to hatch.

## 2. Material and Methods

### 2.1. Collection of Ovaries

Ovaries were collected after the routine ovariohysterectomy surgery of healthy, adult female domestic cats from local veterinary clinics in Krakow. Ovaries were transported to the laboratory in DPBS with streptomycin (100 ug/mL) and penicillin (100 ug/mL) at 4 °C. 

### 2.2. Collection and Classification of Oocytes

Oocyte–cumulus complexes (COC’s) were collected within 1–3 h after ovariohysterectomy by the scarification of the ovarian cortex into a TCM-199 medium with Earle’s salts supplemented with 10% fetal bovine serum (washing medium). Only oocytes with a dark cytoplasm surrounded by compact cumulus cells were assigned for in vitro maturation. 

### 2.3. In Vitro Maturation of Oocytes (IVM)

Selected oocytes were in vitro matured in a BO-IVM (in vitro maturation) medium (IVF-bioscience, Poland) (400 μL) in 4-well dishes. Maturation was carried out for 24 h at 38.5 °C under 5% CO_2_. After IVM, oocytes were denuded of cumulus cells by mechanical pipetting in hyaluronidase (80 IU/mL) for 2 min and then washed in the washing medium described above. 

### 2.4. In Vitro Fertilization (IVF)

The frozen semen used for research was stored in the semen bank of Wrocław University of Environmental and Life Sciences, Department of Reproduction and Clinic of Farm Animals. The semen came from healthy reproducers with known semen quality parameters.

Before the insemination of the oocytes, the semen was thawed at 37 °C for 3 min. After that, the semen was centrifuged with the Sperm Air Medium (400 μL) (Gynemed, Germany) and then incubated at 38.5 °C for 30 min to capacitate the spermatozoa [15]. The capacitated spermatozoa were then selected by using the swim-up method [16]. 

The matured oocytes were inseminated with 5 × 10^5^ capacitated spermatozoa/mL in a BO-IVC (in vitro culture) medium (IVF-bioscience, Poland, Sokolow Podlaski) (400 μL) in 4-well dishes and then incubated together for 16 h at 38 °C under 5% CO_2_. At 16 h post IVF, the spermatozoa were extracted. For each series, 16 randomly selected presumptive zygotes were placed into a Primo-Vision 16-well dish containing the BO-IVC medium. The remainder of the presumptive zygotes were stored separately in a BO-IVC medium (30 μL) in a Petri dish under mineral oil until the first cleavage division for statistical analysis. 

### 2.5. In Vitro Culture (IVC) of Embryos in the Time-Lapse System

A Primo-Vision 16-well dish was prepared and equilibrated for 3 h before the in vitro culture of the embryos. For dish preparation, all 16 wells were filled with the BO-IVC medium and then covered by common 50 μL drops of the BO-IVC medium under 4 mL of mineral oil (Gynemed) (in accordance with the manufacturer’s instructions). Next, the embryos were placed into the microwells, and the loaded dishes were placed into a Primo-Vision microscope located inside the incubator. The embryos were photographed at 10 min intervals. Every other day, out of each 50 μL drop of the BO-IVC medium, 30 μL were removed and replaced by 30 μL of fresh medium in order to maintain stable culture conditions.

### 2.6. Analysis of Frequency of Embryo Characteristics

Presence of a blastocyst cavity.Hatching rate (%).Normal morphology (regular blastomeres and absence of morphological defects).Presence of two or more morphological defects.Presence of one morphological defect.

Examples of the developmental stages of normal embryos and embryos with morphological defects are shown in Table 1.

Morphological defects observed during the study: Cytoplasmic fragmentation (Appendix A)—Portions of cytoplasm surrounded by a cell membrane [4].Cytoplasmic vacuoles (Appendix A)—Cytoplasmic inclusions surrounded by a membrane and filled with fluid [6].Direct cleavage (Appendix A)—Embryo cleavage from 1–3 or 2–3 blastomeres all at once in less than 5 h between divisions [7,9].

## 3. Results

During the study, 435 oocytes were collected from 20 ovaries. Of all the oocytes, 212 were classified and designated for in vitro maturation and in vitro fertilization. Then, 96 presumptive zygotes were randomly selected and observed by using TLM, and 76 of these developed further. Out of the 76 developed embryos, 41 (53.9%) were morphologically normal, and 15 of these (36.6%) reached the blastocyst stage. The embryos with abnormal development were divided into two groups depending on their number of morphological disorders. Of the 35 (46%) abnormally developed embryos, 25 had multiple morphological aberrations and 10 had a single aberration. From abnormally developed embryos, 17 (48.6%) reached the blastocyst stage; six of these had a single aberration, and 11 had multiple aberrations (Table 2). The hatching rate (%) was the highest in the normally cleaving embryos (15.6%), but hatching was also observed within the groups that exhibited a single aberration (6.25%) and multiple aberrations (3.33%) (Table 2).

## 4. Discussion

The present study presents the first results, found by using time lapse monitoring, on the frequency of morphological defects in feline embryos, their competence to reach the blastocyst stage, and their ability to hatch. Until now, all data about cat embryo development have been based on traditional, i.e., discontinuous, observation methods [14,17]. TLM is an effective method for the continuous imaging of the development of each individual embryo in vitro, allowing for the analysis of the morphokinetics, blastomere number, symmetry of cell division, and extent of cytoplasmic fragmentation.

During in vitro culture, embryos remain completely undisturbed, though still under thorough control during the whole period of in vitro development, in the incubator, thus providing the maximum amount of information in order to achieve optimal embryo selection while accurately recording all details of embryo development [4,5]. TLM enables the precise analysis of morphological disorders, including those not observable in traditional ways, e.g., direct cleavage. Embryos with direct cleavage develop better, but the pregnancy rate with such embryos is lower [10,11]. This may be related to incomplete DNA replication and the consequently high frequency of chromosomal abnormalities, as observed in human and bovine embryos [11,18]. The cause of direct cleavage is the formation of tripolar or tetrapolar spindles, which is associated with the abnormal distribution of chromosomes in the blastomeres [18,19]. The most frequently occurring abnormality observed in human IVF embryos is the fragmentation of the cytoplasm [4,8]. It has been shown that the degree of fragmentation is one of the key parameters necessary to determine the quality of embryos. The presence of fragmentation lowers the viability of embryos, lowers implantation rates, and, consequently, reduces pregnancy rates [20,21]. According to published studies [4,8], embryos with fragmentation covering less than 20% of the surface of the cytoplasm have similar implantation results to embryos with normal cytoplasm. Ziebe et al. [4] and Alikani et al. [20] reported that embryos with fragmentation >30–50% have a low potential to implant. However, more than 40% of highly fragmented embryos are chromosomally normal [22], and they can reach the blastocyst stage [23]. Nevertheless, fragmentation >20% is closely related to apoptosis and necrosis [24] and leads to damaged mitochondria in embryos, which is why fragmented embryos are rejected for embryo transfer [25]. 

Cytoplasmic vacuoles are another aberration that are observed during embryo development. Vacuoles are cytoplasmic inclusions that are surrounded by membranes and filled with fluid [6]. They arise as a result of the fusion of preexisting vesicles from the Golgi apparatus or smooth endoplasmic reticulum, or they arise spontaneously [26]. The presence of vacuoles in the cytoplasm could reduce fertilization rates and embryo development [12,13] because the meiotic spindle can be displaced or the cytoskeleton can be disturbed by large vacuoles [27].

In our study, the morphological disorders of cat embryos were observed by using TLM. The embryos were divided into two groups depending on their number of morphological disorders. In the presence of multiple morphological disorders, it was not possible to indicate which of the defects affected the further development of the embryos. Magata and coworkers [3] described abnormal embryo patterns in 46.9% of bovine embryos, a rate that is similar to our results. 

Additionally, in our study, embryos with morphological aberrations had the potential to hatch, but their hatching rate was lower compared to embryos that exhibited normal development. These data are similar to results with bovine [3] and human embryos [2]. 

Due to the critically low population of wild felids, it is important to incorporate new techniques of assisted reproductive techniques (ART). Therefore, the domestic cat was used as a model, because research material from wild felids is too valuable to carry out research on it.

## 5. Conclusions

Embryos with morphological disorders have the potential to reach the blastocyst stage, but they may lack the ability to further develop due to the loss of genetic material, and they may undergo early resorption or miscarriage. Therefore, the use of TLM systems along with ART is extraordinarily helpful for working with extremely valuable research material from wild felids with a critically low population.

## Figures and Tables

**Table 1 animals-10-00003-t001:** Examples of developmental stages of normal embryos and embryos with morphological defects.

Stadium of Development	Embryo Development			
	Normal	Direct Cleavage	Fragmentation of Cytoplasm	Cytoplasmic Vacuoles
Zygote	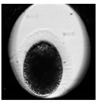	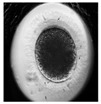	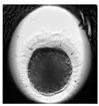	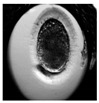
First cleavage	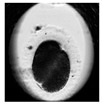	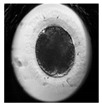	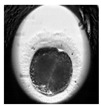	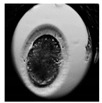
Morula	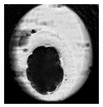	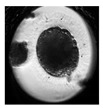	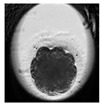	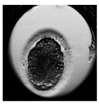
Blastocyst	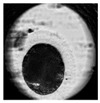	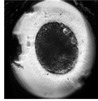	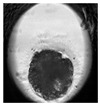	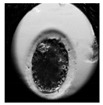
Expanded blastocyst	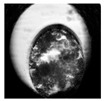	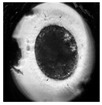	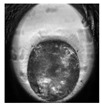	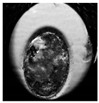

**Table 2 animals-10-00003-t002:** The number of cleavaged embryos and number and hatching rate (%) in normal, multiple or single morphological defects blastocysts.

	Normal	Multi	Single
Number of cleavage embryos	41	25	10
Number of blastocysts	15	11	6
Hatching rate (%)	62.0	13.2	24.8

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
