# Peer review of "Using Time Lapse Monitoring for Determination of Morphological Defect Frequency in Feline Embryos after in Vitro Fertilization (IVF)"

_animals, 2019, doi:10.3390/ani10010003_

Round 1

Reviewer 1 Report

Authors have improved notably the writing of the manuscript.
Nevertheless, I have made several suggestion on the document attached.
I highly apreciate the videos that they provided. They provide a lot of new information, specially for readers with lower notion of cat embryo development. It is not an animal model usually used in publications. However, the manuscript need a figure (table mode as for example 2x5) where the authors show every stage (for example 2 cells, 8 cells, morula, hatching or 4 , 8 , 16 hours...) of a regular embryo development and it would be nice if the can show in paralell the development of an embryo with defects. They should work in this figure and also highly the embryo defects.

Please you should explain much better how did you get the number for the graphs. It seems they are wrong.

Normal development Embryo deffect
2 hours. This is an example the authors should choose the timing they considered the better for their results  
6 hours  
16 hours  
72 hours  

Author Response

Thank you very much for any comments and suggestion for manuscript (Animals-640296). All suggestions have been included in the text or answered. Exact details about the improvement are provided below.

Responds for comments and suggestions into review:

-Reviewer suggested adding a table containing accurate data about embryo development, including: division times etc. The addition of the table misses the purpose of the current manuscript, which is to present the number of developmental disorders of the domestic cat embryos obtained after in vitro fertilization. This is a very interesting topic, but it is necessary to collect more research material and to properly prepare both the introduction and discussion for this subject.

- “Please, you should explain much better how did you get the number for the graphs. It seems they are wrong” - Due to a mistake, incorrect data were included into figure. Figure has been improved.

Responds for comments and suggestions included into attached file:

- “I have serious doubts that capacitation expression should be used here, unless authors give me a reference that half an hour is enough for cats sperm to achieve capacited status” -  The time chosen for the capacitation of frozen feline semen was selected on the basis of many years of experience of the co-authors of the publication, and its positive effect is demonstrated by the ability to fertilize, which is confirmed by the presented results.

-”Please specify the percentage between parenthesis” - Percentage between parenthesis have been changed.

-”Might I be wrong but the percentage doesn’t match with the number the authors said before. For example from the normal embryos:15 form 41 in total reach blastocysts. This means 37% of blastocysts. Please, specify why you only have 15% in the graph. How did you do the calculation?” - Due to the mistake, in the graph have been included number of blastocyst not a percentage. The chart has been improved. Information on the origin of data contained in the chart has also been made more detailed.

-”Please revise the manuscript and normalize styles and size font” - styles nad size font has been corrected.

-”This sentence is confusing. You mean that less of 20% of embryos with cytoplasmic fragmentation achieve implantation? Please rephrase it” - Sentence:” According to published studies [4,7], embryos with cytoplasmic fragmentation less than 20% have similar implantation results to embryos with normal cytoplasm” has been changed for: “according the published studies [4,7], embryos with fragmentation covering less than 20% of the surface of the cytoplasm have similar implantation results to embryos with normal cytoplasm”.

-In line 155 reviewer suggested to changed “are” in sentence “cytoplasmic vacuoles are..” to “is”. Language correctness was checked by native speaker. After that “are” in that sentence has been accepted.

-In sentence, in line 157-158 the missing word has been added.

-Reviewer asked about proper hatching of blastocyst showed on appendix A,B and C. None of them hatched properly.

Reviewer 2 Report

Line 119-120, 124-125: Figures are blurry. Line 129: This study presents the first results…. Line 76-77: with known semen quality parameters. Before insemination of oocytes (Change the size of text). Line 116-117: rate (%) was the highest in (Change the size of text). Line 129-131: feline embryos, their competence to reach the blastocyst stage and ability to hatch using time lapse monitoring (Change the size of text).

Author Response

Thank you very much for any comments and suggestion for manuscript (Animals-640296). All suggestions have been included in the text or answered. Exact details about the improvement are provided below.

-Line 119-120: “Figures are blurry” - Figures has been corrected

-Line 76-77; 116-117; 129-131: “ Change the size of text” - Size of text has been changed.

Round 2

Reviewer 1 Report

Authors amended some of the errors that were detected mainly in the writing part.
However, they omitted the suggestions made regarding the calculation of the ratio. In my modest opinion the figure 1 is wrong. The quality of the figure is so low that I can´t apreciate the number but this graph should be presented as percentages.
Besides, they didn´t include a new figure with pictures of embryos at different stages of development as it was suggested.
As far I concerned, the manuscript doesn´t achieve quality enough to be published in this journal.

Author Response

Dear reviewer, 

Thank you for your sugestions. 

Exact details about the improvement are provided below.

-“Authors amended some of the errors that were detected mainly in the writing part.

However, they omitted the suggestions made regarding the calculation of the ratio. In my modest opinion the figure 1 is wrong. The quality of the figure is so low that I can´t apreciate the number but this graph should be presented as percentages.” The reviewer suggests presenting chart 1 as a percentage. However, we believe that in this form it does not reflect the research problem, therefore it was presented in numerical form. For comparison, in the attachment a graph showing the percentages.As an author, I think that such visualization of results is easier to interpret by other scientists.

-“Besides, they didn´t include a new figure with pictures of embryos at different stages of development as it was suggested.” Table with stages of embryo development has been added.

Round 3

Reviewer 1 Report

Authors improved the manuscript

Author Response

As a respond for editoral office:

Figures has been changed for table 

Citation has been added. 

Tkank you for all revisons